# Nebulized and intravenous enzyme replacement therapy in mice with mucopolysaccharidosis type II

Alex J. Shamoun[1], Gisienne Reis[2], Malaica Ashley[1,2], Anatalia Labilloy[3,4‡],
Leonardo F. Ferreira[1‡*]

1 Department of Orthopaedic Surgery, Duke University School of Medicine, Durham, North Carolina, United States of America, 2 Department of Applied Physiology and Kinesiology, University of Florida, Gainesville, Florida, United States of America, 3 Rare Diseases Unit, Sanofi, Cambridge, Massachusetts, United States of America, 4 Department of Pediatrics, Boston Children's Hospital, Boston, Massachusetts, United States of America

‡ Co-senior authors.
* leonardo.ferreira@duke.edu

## Abstract

Mucopolysaccharidosis Type II is a hereditary lysosomal storage disease characterized by deficiency in the enzyme iduronate 2-sulfatase (IDS). IDS is critical in the breakdown of sulfated glycosaminoglycans and its deficiency leads to an accumulation of these compounds across various tissue types resulting in multi-systemic dysfunction. Intravenous administration of recombinant IDS (idursulfase) substantially improves patients' quality and length of life. However, recombinant IDS delivered intravenously is sequestered in the liver and respiratory failure remains as the leading cause of death for patients independent of idursulfase treatment, which suggests insufficient delivery to the lungs. This study aimed to assess a novel method of idursulfase administration using a nebulizer in combination with intravenous treatment and determine if this combination may improve lung delivery of idursulfase and overall pathology. Whole body IDS knockout mice underwent twelve weeks of intravenous, combination treatment, or vehicle injection and we harvested liver and lungs seven days after the last treatment for assessment of IDS activity, histological markers, and global proteomics for comparison with wild-type mice. Combination treatment increased IDS enzyme activity in the liver but not lungs Proteomics data demonstrated attenuation of key features of the disease in liver (metabolic pathways) and lungs (glycosaminoglycan pathways) with both treatments. Overall, adding nebulized administration of IDS did not lead to sustained increase in enzyme activity in the lungs but caused persistent modifications in glycosaminoglycan degradation pathway suggesting additional benefits to intravenous administration alone.

**Data availability statement:** All relevant data are within the manuscript and its Supporting Information files. The mass spectrometry proteomics data have been deposited to the ProteomeXchange Consortium via the PRIDE partner repository with the dataset identifier PXD054411.

**Funding:** This study was supported by an investigator-initiated grant from Takeda Pharmaceuticals (IIR-USA-001523) to AL

**Competing interests:** The authors have declared that no competing interests exist.

## Introduction

Mucopolysaccharidosis Type II (MPS II) is an X-linked lysosomal storage disease characterized by deficiency in the lysosomal hydrolase iduronate 2-sulfatase (IDS), a critical enzyme in the breakdown of the glycosaminoglycans (GAGs) heparan sulfate and dermatan sulfate [1]. The resulting GAG accumulation increases lysosome size and number and detrimentally affects cellular adhesion, intracellular transport, and inflammatory regulation, leading to the pervasive multisystemic dysfunction responsible for the wide scope of the disease symptoms [2]. Clinical manifestations of the disease include elevated urinary GAGs, coarse facial features, organomegaly, growth delay, cardiac and respiratory dysfunction [1,3,4], central nervous system impairment [5,6], restricted joint mobility [7], and a myriad others representative of the multisystemic dysfunction inherent to the disease. MPS II is a relatively rare diagnosis, with an incidence rate varying regionally from 0.38 to 1.09 per 100,000 live births [1]. MPS II is not associated with a single genotype, but with a wide variety of mutations of the IDS gene. Severe mutations resulting in the complete lack of functional enzyme seem to result in the severe phenotype, while single amino acid point mutations have been associated with various phenotypes ranging from attenuated to severe [8].

Idursulfase (ELAPRASE®, Takeda, Lexington, MA), a purified recombinant form of IDS produced in human cell lines, administered intravenously at a weekly dosage of 0.5 mg/kg in patients with MPS II demonstrated significant improvements in multiple markers of cardiorespiratory health compared to the placebo group, including 6-minute-walk distance, predicted forced vital capacity, and absolute forced vital capacity [9] resulting in enzyme replacement therapy becoming an FDA-approved treatment for MPS II [10]. A subsequent analysis of a global multicenter registry for MPS II patient data showed that idursulfase treatment increases survival and lowers the risk of death in patients [11]. However, a study in mice showed tissue heterogeneity of idursulfase, with liver uptake accounting for 30–40% of the dose and 60- to 80-fold higher uptake than kidneys, heart, or spleen after intravenous injection [12]. Accordingly, respiratory failure remains the leading cause of death for both treated and untreated patients with MPS II, despite improvements in multiple respiratory markers with enzyme replacement therapy (ERT) [3,9,11], showing that new approaches are needed to target the lungs and respiratory pathology in patients receiving enzyme replacement therapy.

The goal of the present study was to test a novel method of ERT administration via nebulization in combination with intravenous administration ('standard of care' for enzyme replacement therapy) to enhance lung delivery of idursulfase and improve pulmonary pathology. We defined liver and lung IDS activity, histology, and proteomics following treatment in a mouse model of MPS II. Our hypothesis was that nebulized idursulfase would enhance IDS activity and improve the MPS II pathology in the lungs compared to intravenous delivery alone.

## Methods

### Animals and intervention

The procedures in this study were approved by the University of Florida Institutional Animal Care and Use Committee (Protocol # 2020112244). Male wildtype C57BL/6N

and IDS-KO (024744, B6N.Cg-Ids^tm1Muen/J; Jackson Laboratory, Bar Harbor, ME) mice were used in this study. The mice were housed under a 12h:12h light-dark cycle with access to standard chow and water ad libitum. Male mice were divided into four groups: 1) Wildtype mice receiving saline intravenous (IV) and nebulized (*WT*, *n* = 8); 2) IDS-KO group receiving saline IV and nebulized (*KO*, *n* = 5); 3) IDS-KO receiving 1 mg/kg idursulfase IV (*KO-IV*, *n* = 5); 4) IDS-KO receiving 1 mg/kg idursulfase IV and nebulized idursulfase (*KO-NEB*, *n* = 5). The nebulized idursulfase was delivered as 167 μL of 2 mg/mL. Treatment was administered 1x/week starting at eight weeks of age, and terminal experiments were conducted 12 weeks after the onset of the intervention, seven days after the final administration of the treatment. Mice were deeply anesthetized with isoflurane (5% induction, 2–3% maintenance) prior to undergoing a laparotomy to collect liver tissue and thoracotomy for euthanasia via removal of the heart and collection of lung tissue. Samples were rinsed in PBS, diced with surgical scissors, and separated into tubes for flash freezing in liquid $N_2$ and storage at −80 °C or fixation in 10% formalin at 4 °C for 12 hours.

### IDS activity assay

IDS activity was measured in liver and lung samples using an adaptation of a previously described fluorescence-based enzymatic activity assay [13]. Frozen tissue samples were hand-homogenized in a Kimble® Kontes® Duall® Tissue Grinder on ice in Tris Buffer (10 mM Tris, pH 7.5). Homogenates were transferred to a microcentrifuge tube and sonicated 3 seconds at 30% amplitude (Fisher Scientific 40:0.15:4C control panel; model #4C15 sonicator). Samples were centrifuged at 10,000 *g* for 15 minutes at 4 °C. Supernatant was transferred to a new microcentrifuge tube and the pellet discarded. Total protein concentration of the supernatant was determined using a NanoDrop™ UV-Vis spectrophotometer (ThermoScientific). Samples were diluted to a uniform protein concentration (~6 μg/μl) and 120 μg of total protein was loaded in each well for the assay.

The IDS activity assay uses the fluorometric-tagged substrate 4-methylumbelliferyl α-L-iduronide-2-sulfate (4-MUS) (9001600; Cayman Chemical, Ann Arbor, MI). In Step 1, IDS catalyzes the hydrolysis of 4-MUS to 4-methylumbelliferyl α-L-iduronide (MUBI) [13]. The 4-MUS solution (0.416 mM 4-MUS, 100 mM sodium acetate, 10 mM lead (II) acetate, pH 5.0) was prepared in advance and aliquots stored at −20 °C until use. In a black 96-well plate on ice, sample was combined with 20 μL of the 4-MUS solution. Additional wells containing 10 μL of the Tris Buffer with 20 μL of the 4-MUS solution in triplicate were treated the same as the sample-containing wells throughout the protocol, to which the 4-MU standards would be added at the end to create the 4-MU standard curve. The lidded plate was sealed with parafilm, covered in foil, and incubated in a shaker at 37 °C, 200 RPM for one hour, and immediately placed on ice for Step 2.

In Step 2, MUBI produced in Step 1 is further hydrolyzed to the fluorescent compound 4-methylumbelliferone (4-MU) by α-L-iduronidase (IDUA) [13]. A stop solution for the first reaction containing recombinant IDUA (FI179383; Biosynth) (22 μg/mL in 400 mM sodium phosphate dibasic, 200 mM sodium citrate, pH 4.5) was prepared and 45 μL was added per well. The plate was again covered in foil and incubated in shaker at 37 °C, 200 RPM for four hours. A standard curve was prepared with 4-MU (HY-N0187; MedChemExpress) prepared in Carbonate Stop Buffer (500 mM sodium carbonate, 500 mM sodium bicarbonate, 0.025% v/v Triton X-100, pH 10.7) and kept on ice. When the second incubation was completed, 200 μL Carbonate Stop Buffer was added to each sample well, and 200 μL of the appropriate standard to each standard well. Fluorescence was measured on a Molecular Devices SpectraMax® i3x microplate reader at 365 nm excitation, 445 nm emission.

### Sample preparation for global proteomics

Lung and liver samples for proteomics analysis were flash-frozen in liquid $N_2$ and stored at −80 °C until processed. Sample preparation and Nano-LC/MS/MS were completed by the University of Florida Proteomics and Mass Spectroscopy Core. The EasyPep™ MS Sample Prep Kit (Thermo Fisher Scientific) was used for protein digestion and extraction. The sample volume to contain 100 μg total protein was determined using total protein analysis on a Quibit and taken for

digestion. The samples were then digested using the sequencing grade trypsin/lys C rapid digestion kit (Promega) according to the manufacturer's recommended protocol. Digestion buffer was added to samples in a 3:1 volume ratio. Digested samples were incubated at 56 °C with 1 μL of 0.1 M DTT in 100 mM ammonium bicarbonate for 30 minutes, followed by adding 0.54 μL of 55 mM Iodoacetamide in 100 mM ammonium bicarbonate for a 30-minute incubation in the dark at room temperature. The trypsin/lys C was freshly prepared at a concentration of 1 μg/μl in the rapid digestion buffer. 1 μL of this enzyme was added and the samples were incubated at 70 °C for 1 hour. The digestion process was halted by adding 0.5% TFA.

### Nano-LC/MS/MS

The analysis was performed immediately upon the completion of sample preparation to ensure high sample quality. Nano-liquid chromatography tandem mass spectrometry (Nano-LC/MS/MS) was performed using a Q Exactive HF Orbitrap mass spectrometer with an EASY Spray nanospray source operated in positive ion mode and an UltiMate™ 3000 RSLCnano system with a capillary temperature of 200 °C and a spray voltage of 1.5 kV (Thermo Scientific). Mobile phase A and mobile phase B were 0.1% formic acid in $H_2O$ and acetonitrile, respectively. The mobile phase A for the loading pump was 0.1% trifluoroacetic acid in $H_2O$. Then, 5 μL of the processed sample was injected on to a PharmaFluidics μPAC™ C18 trapping column at a flow rate of 10 μL/mL for 3 minutes prior to a wash with 1% B. Chromatographic separation was conducted using a PharmaFluidics 50 cm μPAC™ at 40 °C. The flow rate was maintained at 750 nL/min for 15 minutes and at 300 nL/min thereafter. A 1% B to 20% B gradient for 100 minutes followed by 45% B for 20 minutes were used to elute peptides off the column into the Q Exactive HF Orbitrap system.

The mass spectrometer was run with a scan sequence based on the original TopTen™ method with full scan from 375 to 1575 Da at 60,000 resolution and MS/MS scan at 15,000 resolution. Product ion spectra were generated to determine amino acid sequence in consecutive scans of the 15 most abundant peaks in the spectrum. AGC target ion number was 300,000 with 50 ms injection time and 20,000 with 55 ms injection time for full scan and MS/MS mode, respectively, and (N)CE/ stepped NCE was set to 28. The isolation window was set to 4 $m/z$. MS/MS analysis excluded singly charged ions. Internal lock mass was evaluated using a Siloxane background peak at 445.12003.

### Histology

The tissue samples were fixed in 10% formalin and embedded in paraffin. Samples were sectioned at 4 μm thickness and stained with Alcian Blue and Masson's Trichrome stains by the Molecular Pathology Core at the University of Florida. The Alcian Blue kit was purchased from Abcam (ab150662, MA). Slides were deparaffinized and rehydrated, then placed in acetic acid solution for 3 min, prior to incubation with Alcian Blue (pH 2.5) for 30 min. Slides were then placed neutral fast red for 5 min. After rinsing and dehydrating, the slides were mounted with cytoseal. Masson's Trichrome kit was purchased from Richard Allan Scientific (FS87019, Fisher Scientific). The staining was completed according to the manufacturer recommended protocol. Briefly, slides were deparaffinized and rehydrated, then placed in Weigert's Iron Hematoxylin for 5 min, followed by incubation in Biebrich Scarlet/ Acid Fuchsin solution for 7 min and differentiation in phosphomolybdic/ phosphotungstic acid solution for 5 min. Slides were then incubated in Aniline Blue solution for 7 min and then in acetic acid solution for 1 min. After rinsing and dehydrating, the slides were mounted with cytoseal.

Histology slides were analyzed using the Fiji image analysis software [14]. For both the Masson's Trichrome and the Alcian Blue stains, the color threshold function was used to quantify the blue-stained pixels in whole-section images. For the Masson's Trichrome stain, the blue was selected to quantify the Aniline Blue dye, a component of the Masson's Trichrome mixture indicative of collagen-rich connective tissue [15]. The Alcian Blue stain selectively binds to acidic glycoproteins [16]. The parameters for the color threshold function were set according to an optimization conducted with wild-type samples. For the Masson's Trichrome stain, the color threshold parameters were set as [lower bound, upper bound]:

hue [132, 196], saturation [0, 255], brightness [80, 255]. For the Alcian Blue stain, these parameters were set as: hue [110, 190], saturation [15, 255], brightness [80, 255]. The blue pixel area was divided by the total pixel area of the section for normalization. Images were converted to 8-bit grayscale and the threshold function was used to calculate total pixels, set to [0, 220] for Masson's Trichrome stained images and [0, 230] for Alcian Blue stained images.

### Statistical analysis

Statistical analysis for the IDS activity assay and the histological stain quantifications were conducted using JMP Pro (JMP Statistical Discovery, Cary, NC; Version 17.2.0). Data were assumed to be non-parametric due to low sample size and groups compared using the Kruskal-Wallis test. Statistical significance was determined when $p < 0.05$. When statistically significant differences were found, post hoc comparisons were conducted using the Wilcoxon Each Pair method.

The MS/MS spectra were analyzed using Sequest (Thermo Fisher Scientific, San Jose, CA; version IseNode in Proteome Discoverer 3.0.1.27). Sequest was set up to search the mouse database and assume trypsin digestion. Tolerances were set to a 0.020 Da fragment ion mass and a 10.0 ppm parent ion. Cabamidomethylation of cysteine was set as a fixed modification. Met-loss, met-loss with acetylation, oxidation, and n-terminal acetylation of methionine were set as variable modifications. Precursor ion intensity label-free quantitation was performed using Proteome Discoverer (Thermo Fisher Scientific, San Jose, CA; version 2.4.0.305). Each combination of two groups was compared using the non-nested study factor and normalization using all peptides. Protein abundances were calculated by summation of peptide group abundances. A 100% cutoff was used to analyze data such that protein abundances were considered only for proteins that were detected in all samples in at least one group. The $p$-values were calculated with Fisher's exact test (pairwise ratio-based) using low-intensity resampling value imputation, and adjusted $p$-values were calculated using Benjamini-Hochberg. Proteins were considered statistically significant where the adjusted $p$-value was less than 0.05. Differentially expressed proteins were defined based on an arbitrary 1.5-fold change and adjusted p-value < 0.05. For enrichment and network analysis we considered upregulated proteins where $\log_2$(abundance ratio) > 0.585 and downregulated proteins were defined as those where $\log_2$(abundance ratio) < 0.585 for pathway enrichment analysis, conducted using ShinyGO 0.80 [17], searching the KEGG pathway database with a false discovery rate cutoff of 0.05. The top 20 differentially expressed KEGG pathways filtered by fold enrichment were identified and are reported here. KEGG Pathways of potential pathological relevance that emerged from liver and lung analysis in all three groups were selected arbitrarily for further analysis of protein interaction network (StringDB). The outputs from proteomics analyses are shown in the results and supplemental files. The mass spectrometry proteomics data have been deposited to the ProteomeXchange Consortium via the PRIDE partner repository with the dataset identifier PXD054411 and is publicly available.

## Results

### IDS activity

In the *KO-IV* group, IDS activity in liver samples increased compared to the *KO* group ($p = 0.012$) and not statistically different from the *WT* group ($p = 0.144$). In liver samples from *KO-NEB* mice, IDS activity was higher than the *KO* group ($p = 0.012$), 203% higher than the *KO-IV* group ($p = 0.012$), and 86% higher than the *WT* group ($p = 0.022$) (Fig 1A). In lung samples, both treated knockout groups had lower IDS activity than the *WT* group (*KO-IV*: $p = 0.037$; *KO-NEB*: $p = 0.012$), and their measured IDS activities were not significantly different from each other ($p = 0.531$) (Fig 1B).

### Histology

There was no difference in fibrosis for both lungs (Fig 2A) and liver (Fig 2B) sections. The percentage area stained for alcian blue (glycosaminoglycans) in the lung was not statistically different among groups but was increased in the liver for *KO* ($p = 0.023$) and *KO-IV* ($p = 0.007$), but not *KO-NEB* compared to the *WT*.

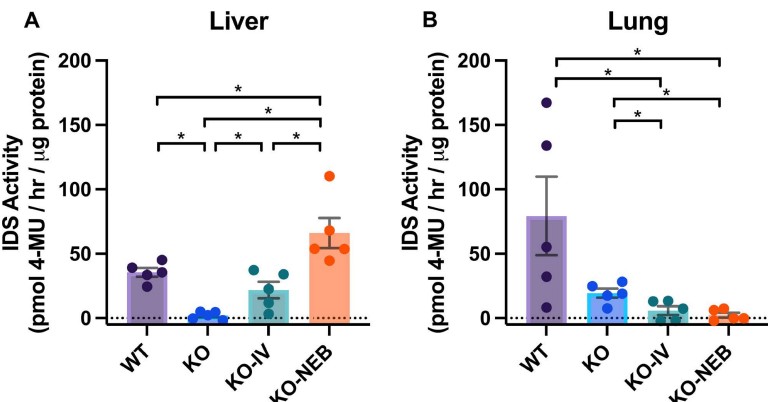

**Fig 1. Administration of recombinant IDS caused persistent increase in liver but not lung enzyme activity.** IDS activity measured in mouse liver (A) and lung (B) samples measured by fluorescence and normalized to total protein content.

## Proteomics analysis

For comparison of proteomic results, we used the *KO* group as reference to minimize the number of comparisons and FDR. All proteomics data output from Proteome Discoverer generated using the criteria established for comparisons are shown in the supplemental files (Supplemental Information). Volcano plots highlight the distribution based on adjusted p-value and fold change for proteins found in all samples from both groups (Fig 3).

Regarding the liver, there were 730 significant DEGs in the *WT* group, 786 in the *KO-IV* group, and 878 in the *KO-NEB* group compared to KO group (Table 1). Metabolic pathways had the largest number of proteins showing upregulation and downregulation in all comparisons (Fig 4). Noticeably, KEGG Ribosome pathway was upregulated and downregulated in KO vs WT and KO vs KO-IV, but only downregulated in KO vs KO-NEB (Fig 4). Protein network and interaction analysis for upregulated and downregulated KEGG metabolic pathways showed similar clusters for all comparisons (Figs 5–6). The KO/KO-IV comparison showed the highest number of proteins in the upregulated metabolic pathway (KO/WT = 113, KO/KO-IV = 404, KO/KO-NEB = 142; Fig 5). These findings are consistent with higher liver IDS activity in KO-NEB vs KO-IV. The KO/KO-NEB comparison showed the highest number of proteins in the downregulated metabolic pathway (KO/WT = 82, KO/KO-IV = 80, KO/KO-NEB = 105; Fig 6). Overall, these findings are consistent with higher liver IDS activity in KO-NEB vs KO-IV.

Regarding the lungs, there were 927 significantly changed proteins in the *WT* group, 941 in the *KO-IV* group, and 954 in the *KO-NEB* group compared to *KO* group (Table 2). Out of the KEGG pathways upregulated in the *KO* group, the glycosaminoglycan degradation pathway had the highest or second-highest fold enrichment compared to *WT, KO-IV*, and *KO-NEB* (Fig 7). Downregulated pathways in the *KO* vs *WT, KO-IV, and KO-NEB* included many associated with various disease states (Fig 7).

## Discussion

The principal findings of this study are that treatment of IDS-KO mice with idursulfase by combination of intravenous and nebulizer administration augmented long-term retention of IDS enzyme activity and prevented the increase in histology GAG levels in the liver but not lung. However, proteomics data revealed a molecular signature on the lung consistent with greater benefits when nebulized administration was combined to intravenous delivery compared to intravenous alone. To our knowledge, this is the first examination of proteomic profiling of MPS II disease and ERT in rodents and equivalent data in humans is not available in the literature or public databases.

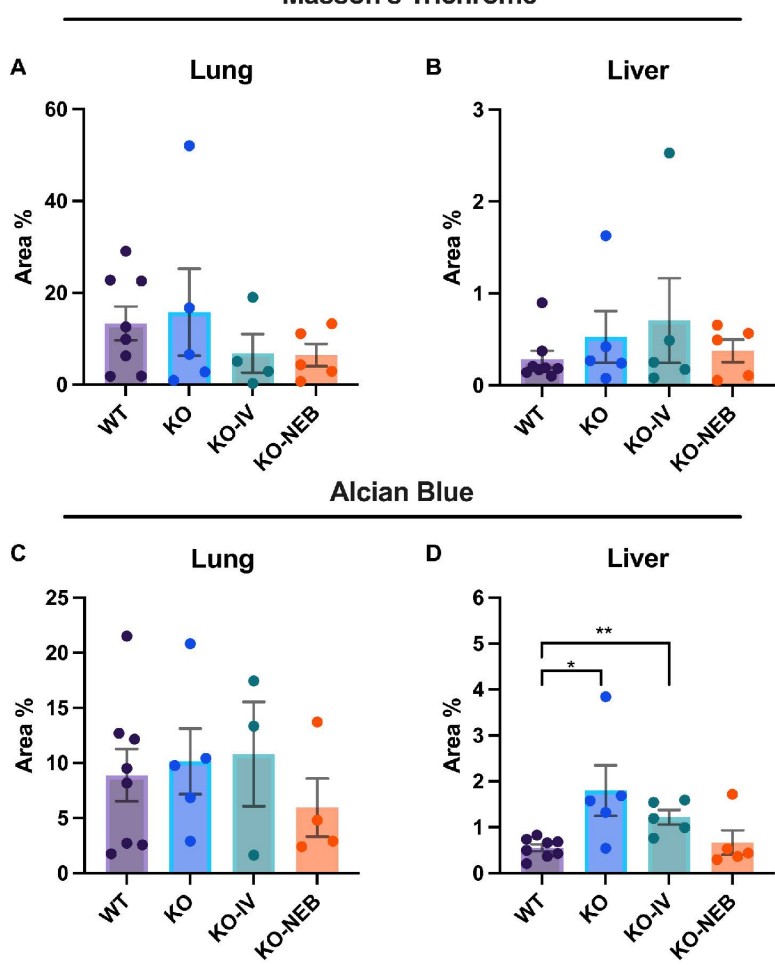

**Fig 2. Quantification of markers of fibrosis analyzed by Masson's Trichrome and glycosaminoglycans analyzed by Alcian Blue staining in lung and liver.** Data shows area stained with Masson's Trichrome or Alcian Blue expressed as a percentage of total tissue section area for each animal. * ($p = 0.023$) and ** ($p = 0.007$) groups compared to the WT group.

Treatment with intravenous administration fully rescued and combination further augmented IDS activity in the liver, which implies that the recombinant enzyme administered by nebulization entered the circulation and was taken up and retained in the liver. The higher liver enzyme activity with nebulization prevented the increase in glycosaminoglycan in histological section compared to controls. Moreover, proteins from a range of metabolic pathways were differentially expressed with deficiency and treatment. The greatest change was evident in upregulation of proteins in the KEGG metabolic pathway of KO vs KO-IV group. Overall, IDS KO resulted in upregulation of multiple pathways broadly involved in carbohydrate metabolism, while some pathways involved in lipid metabolism were downregulated. These findings are consistent with metabolic shift from lipid to carbohydrate metabolism in the liver that is akin to that seen in the heart with cardiomyopathies. The 'Peroxisomes' pathway was downregulated in KO vs WT and KO-NEB, but not when compared to KO-IV. These data suggest that the absence of IDS in MPS-II decreases peroxisomes and ERT delivery via intravenous injection and nebulizer reached liver doses that restored abundance of proteins within peroxisomes and might have contributed to enhancement of hepatic lipid metabolism and diminish liver pathological features in MPS-II.

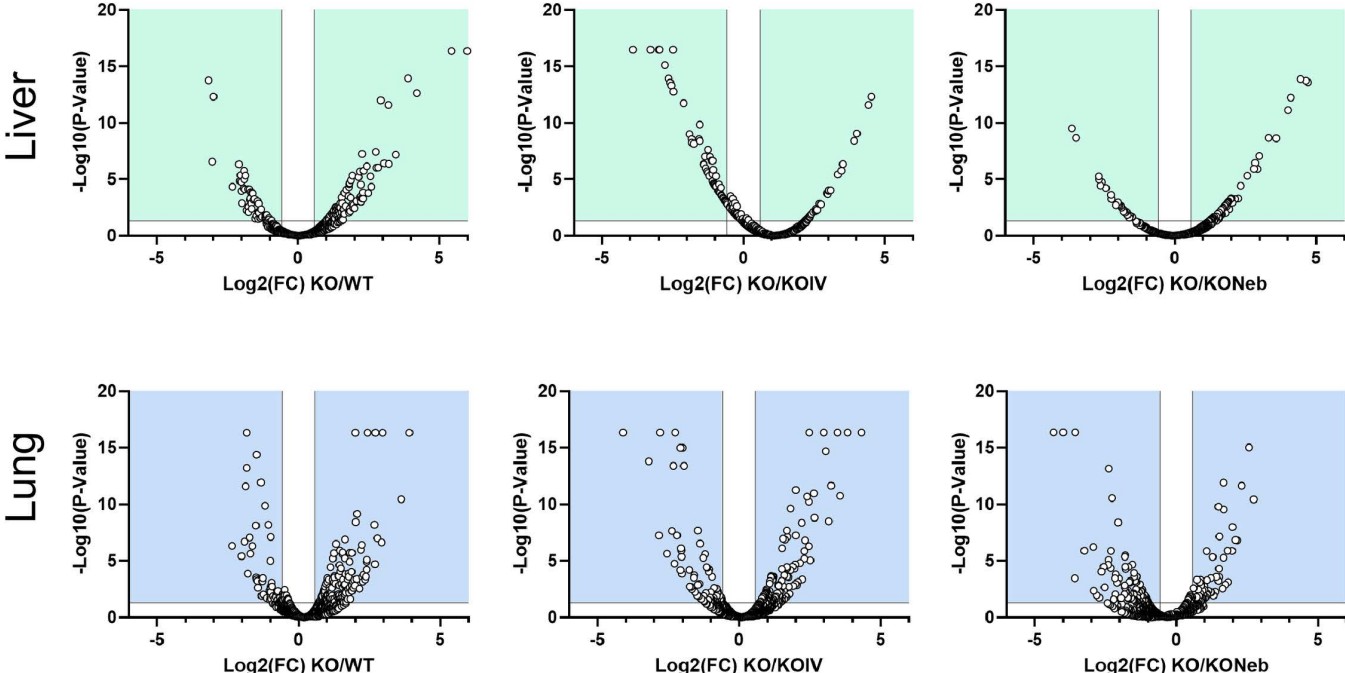

**Fig 3. Volcano plots of proteomics data.** Data shows distribution based on log2 for fold change and -log10(P-value) for proteins present in all samples from both groups. Proteins in the upper left and right corners meet criteria for fold change and adjusted p-value as demarcated by vertical and horizontal lines, respectively.

**Table 1. Liver proteomics outcome.** Data show total differentially expressed proteins defined as $|\log_2(\text{abundance ratio})| \geq 0.585$ (fold change > 1.5) and Benjamini-Hochberg adjusted *p*-value ≤ 0.05. UP and DOWN are number of proteins increased and decreased in KO vs each group.

| KO vs Group | Total Fold Change > 1.5 | *Total FC > 1.5 & padj. ≤ 0.05* | UP | DOWN |
|---|---|---|---|---|
| WT | 1090 | 730 | 400 | 330 |
| KO-IV | 2293 | 786 | 425 | 361 |
| KO-NEB | 1327 | 878 | 734 | 144 |

The detection of IDS activity from enzyme replacement therapy depends on timing of measurements relative to administration. IDS activity detected in each tissue of IDS-KO mice depends on both the acute uptake of the recombinant enzyme and the retention of active enzyme. Acute uptake of IDS is mediated by insulin-like growth factor II/cation-independent mannose 6-phosphate receptor and a critical step as exogenous lysosomal enzymes have been found to be rapidly sequestered from circulation by the liver and spleen [18]. RNA transcriptome data indicates relatively high expression of insulin-like growth factor II receptor in mouse lung [19]. Therefore, high initial uptake in the lung is expected, especially with the direct administration by nebulizer in the combination treatment. In a previous pilot study [20], we found that intravenous injection of IDS caused a similar increase in IDS activity in the lungs and liver of IDS KO mice when measurements were performed 2 hours post-injection. Therefore, it is reasonable to expect that the nebulized delivery increased lung IDS activity acutely and the increase exceeded that elicited by an intravenous injection such that lungs were exposed to a large percentage of the exogenous enzyme lungs initially. However, measurements performed within 2 hours post-injection might reflect circulating IDS rather than tissue uptake. Our current measurements performed seven

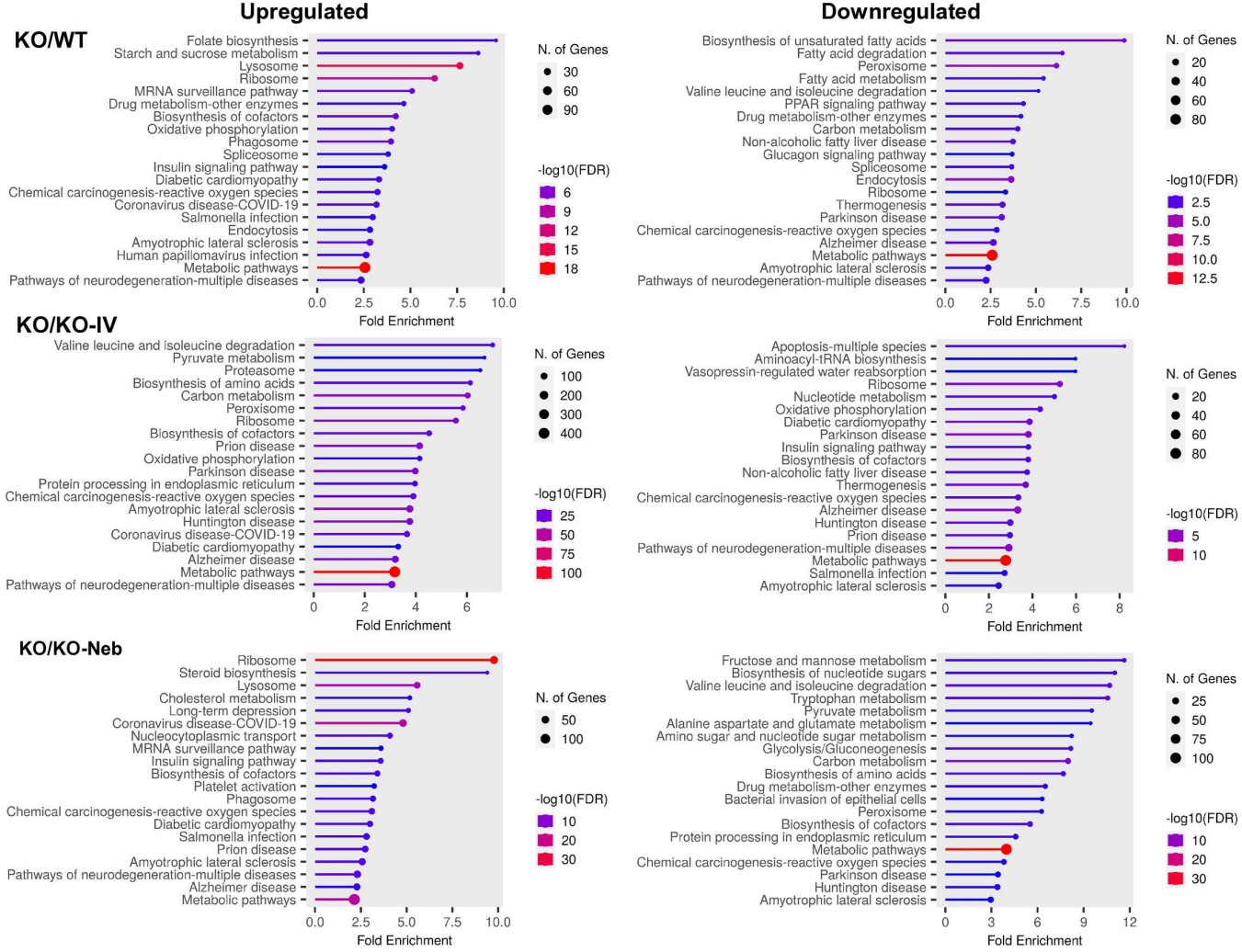

**Fig 4. KEGG pathways diagrams for liver proteomics.** Data show top 20 pathways upregulated and downregulated based on fold enrichment for KO vs WT, KO-IV, and KO-NEB. Hierarchical pathways defined according to fold enrichment with symbol sizes and colors indicating number of genes and -log10 of false discovery rate (FDR).

days after ERT showed that IDS activity in the lungs did not increase with either IV treatment alone or the combination treatment with respect to the *KO* group. Thus, our observations indicate that exogenous IDS, presumably available to the lungs acutely, were not detectable in the lung seven days post-treatment and were retained mostly in the liver. However, a previous study has shown ERT via lentiviral vector increases IDS activity in the long-term lungs [21]. We also did not detect changes in our histology markers of GAG or fibrosis in 5-month-old mice, but these were evident in 10-month-old mice [22]. Despite the lack of detectable IDS activity and histological changes in the lungs under the conditions of the study, the proteomics analysis suggest that there were effects of treatment in the lung proteome. A multitude of proteins were differentially expressed in both treatment groups compared to the untreated *KO* group (Table 2) and various pathways associated with disease states and metabolism differed in expression (Figs 3, 7–8). Interestingly, the glycosaminoglycan degradation KEGG pathway was upregulated in *KO* control vs WT, KO-IV, and KO-NEB. This shows that there are disease and treatment effects on the tissue even though we could not detect elevated enzyme activity and a report in the

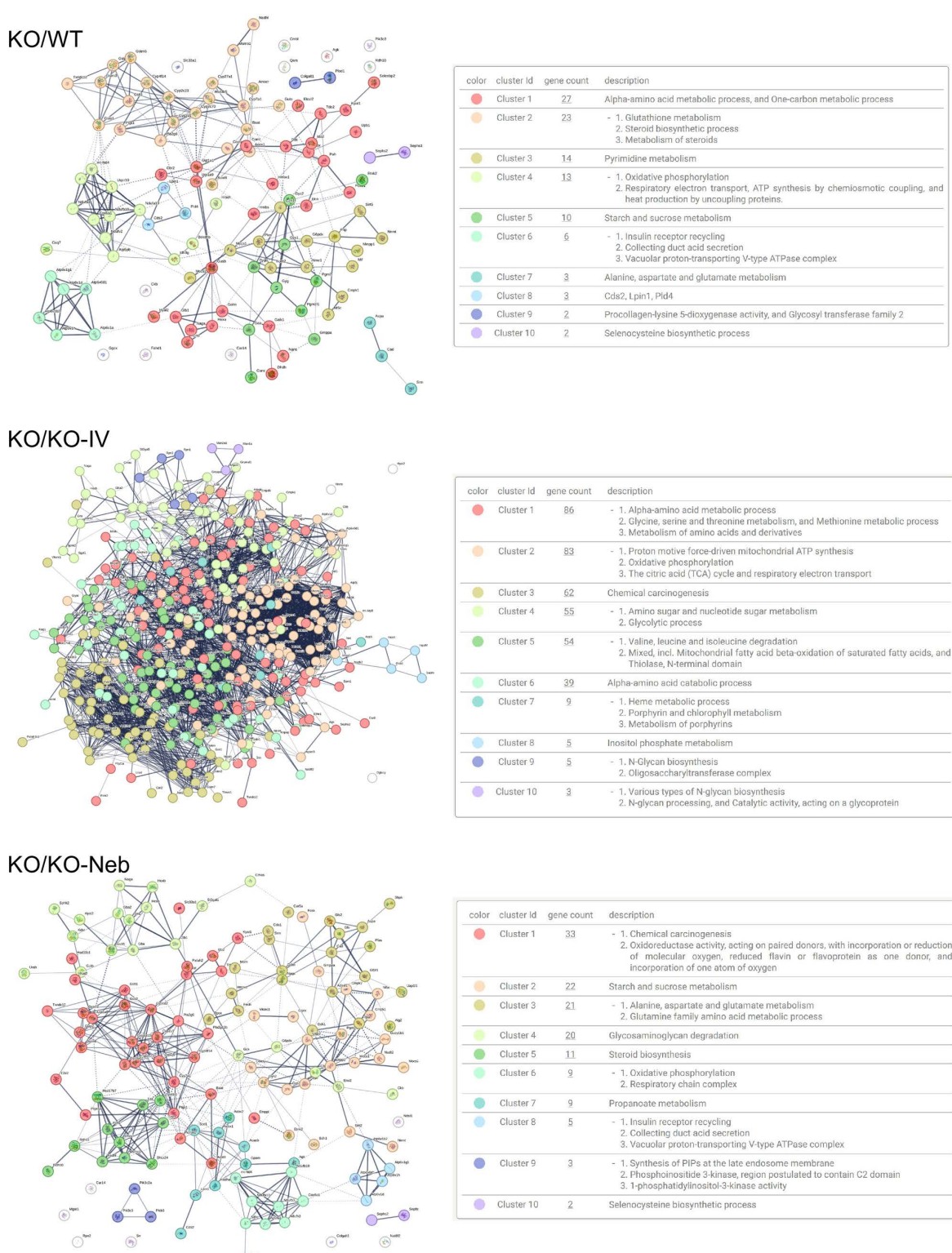

**Fig 5. Protein network and clusters for KEGG metabolic pathway (liver proteomics) uregulated in KO vs. WT or treated groups.** Images are from metabolic pathways upregulated in KO vs WT, KO-IV, and KO-NEB. Network edges mean confidence of interaction and line thickness indicates the strength of data support. Right panels show clusters identified within each network and number of proteins ('gene count') in each cluster.

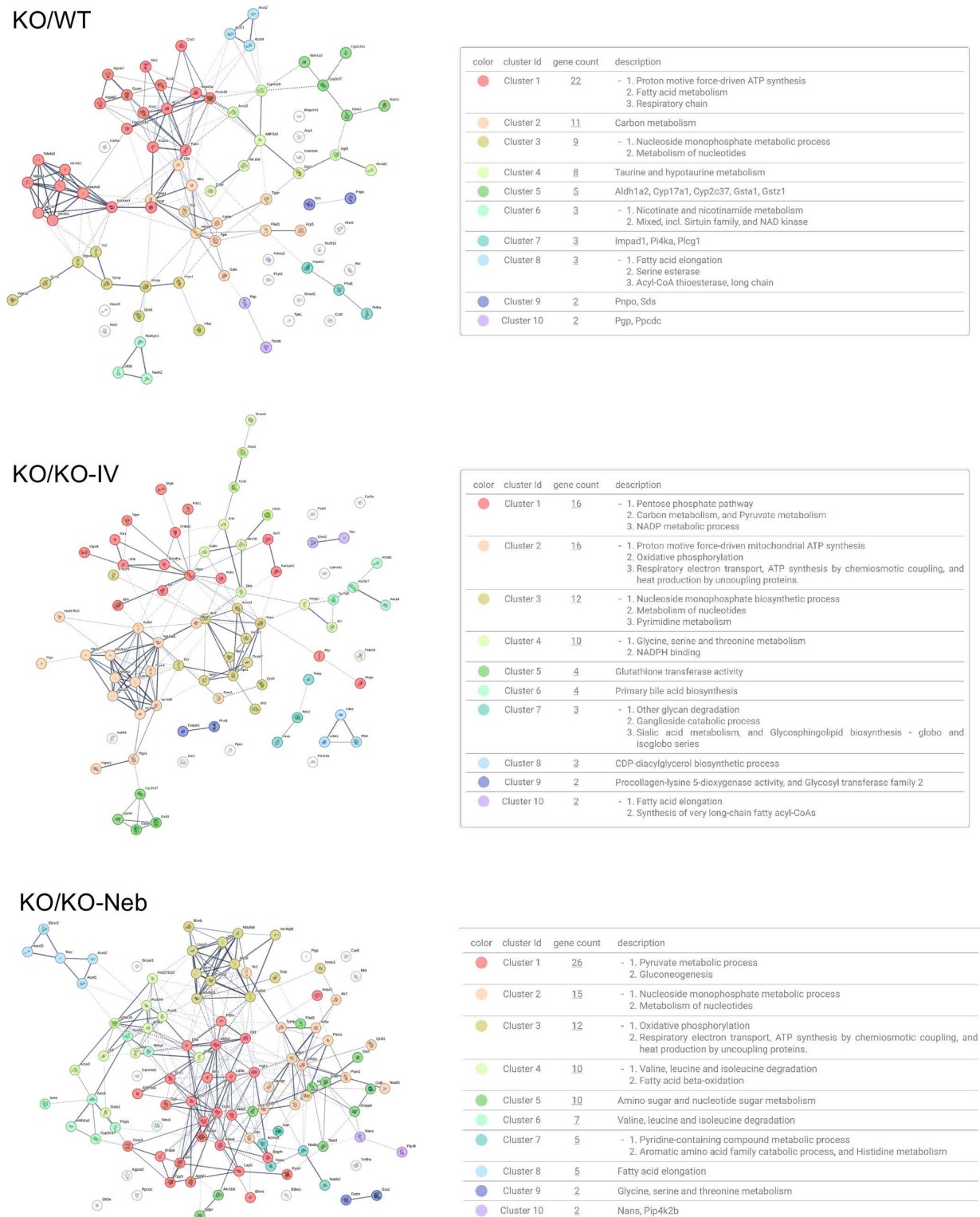

**Fig 6. Protein network and clusters for KEGG metabolic pathway (liver proteomics) downregulated in KO vs. WT or treated groups.** Network edges mean confidence of interaction and line thickness indicates the strength of data support. Right panels show clusters identified within each network and number of proteins ('gene count') in each cluster.

**Table 2. Lung proteomics outcome.** Data show total differentially expressed proteins defined as $\left|\log_2(\text{abundance ratio})\right| \geq 0.585$ (fold change > 1.5) and Benjamini-Hochberg adjusted $p$-value ≤ 0.05. UP and DOWN are number of proteins increased and decreased in KO vs each group.

| KO vs Group | Proteins Fold Change > 1.5 | Proteins FC > 1.5 & padj. ≤ 0.05 | UP | DOWN |
|---|---|---|---|---|
| WT | 1343 | 927 | 464 | 463 |
| KO-IV | 1260 | 941 | 523 | 418 |
| KO-NEB | 1820 | 954 | 556 | 398 |

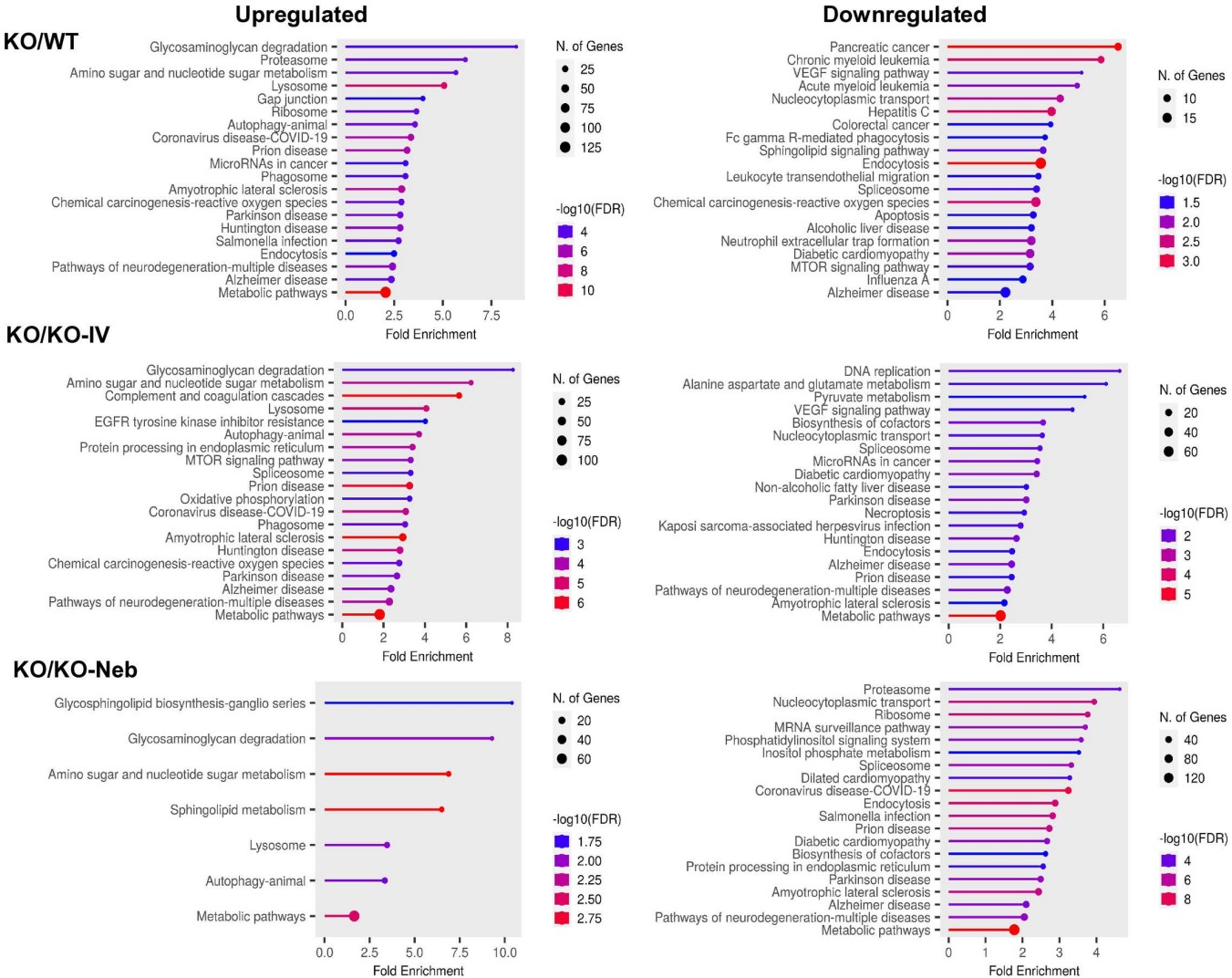

**Fig 7. KEGG pathways diagrams for lung proteomics.** Data show top 20 pathways upregulated and downregulated based on fold enrichment for KO vs WT, KO-IV, and KO-NEB. Hierarchical pathways defined according to fold enrichment with symbol sizes and colors indicating number of genes and -log10 of false discovery rate (FDR).

literature is consistent with short-term pulmonary improvements in an infant receiving ERT [23]. Individual protein analysis showed higher abundance of various enzymes downstream of IDS (Fig 8). The upregulation of enzymes downstream of IDS is consistent with a compensatory mechanism in response to low levels of products from glycosaminoglycan metabolism, which is attenuated by enzyme replacement therapy and appeared to undergo greater attenuation with the addition of nebulized therapy. Overall, the data suggests that exogenous IDS becomes available to the lungs acutely, but most of the long-term enzyme retention occurs in the liver. Regardless, ERT confers some improvement to the lung proteome within 12 weeks of treatment but changes in lung histology and function might require longer duration of treatment or more frequent dosing.

A potential concern with repeated dosing of exogenous IDS is the development of infusion-associated reactions, which have an allergenic nature and include bronchoconstriction [24]. Although rare, adverse reactions are a concern and nebulization could exacerbate bronchoconstriction. There are no data in the literature to suggest that nebulized IDS per se would cause bronchoconstriction when applied in a solution with adequate osmolality. Mice in our study did not show signs of respiratory distress or behavioral discomfort when exposed to nebulized IDS. Future studies would have to test airway reactivity to IDS and pilot studies in patients might consider combining nebulized IDS with bronchodilators as precaution, which could even enhance benefits of nebulization when combined with lung volume recruitment strategies, e.g., positive end-expiratory pressure breathing. Importantly, nebulized ERT might be a feasible approach to bypass limitations of IDS crossing the blood brain barrier and reaching the central nervous system through axonal transport via the olfactory and trigeminal nerves [25–28]. However, these topics were beyond the scope of our investigation. In general, our study suggests that nebulized IDS might be a safe, less invasive, and effective approach for ERT with potential to attenuate lung pathology in patients with MPS II disease.

## Limitations

Limitations in our study include the timing of measurements post-treatment, unequal and undetermined doses of IDS replacement with nebulization, and inability to detect fibrosis. The completion of experiments seven days after the final administration of treatment was aimed to allow investigation of the chronic retention of enzyme activity and assessment of long-term changes in the histological and proteomic properties of the tissue but precludes resolution of acute delivery and

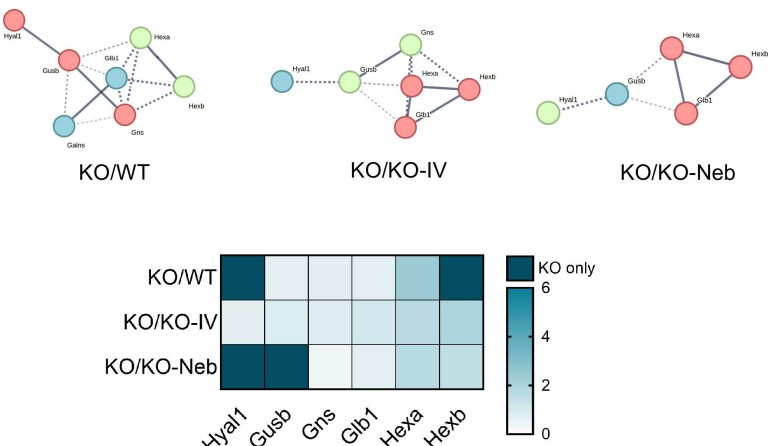

**Fig 8. Protein network for KEGG glycosaminoglycan degradation pathway (lung proteomics) and heatmap of proteins upregulated in KO vs. WT or treated groups.** Network edges show confidence of interaction and line thickness indicates the strength of data support. Heatmap shows proteins from pathway analysis. Color gradient indicates fold change for each comparison. Darkest color indicates proteins present only in KO vs comparison group.

effects of the replacement enzyme. Moreover, we cannot determine the impact of route vs dose of administration when adding nebulized therapy. Proper dosage of the combination therapy is a challenge in this study design because, while a specific dosage can be delivered to the nebulizer chamber, we could not determine the exact amount of enzyme inhaled by the mice. In this setting, adding the amount supplied via nebulizer to the intravenous dose would also result in unequal and possibly excessively high doses. Therefore, we opted to not match the IV dose to the amount nebulized in the chamber. Lastly, despite previous findings of fibrosis in MPS II [29], the Masson's trichrome staining did not detect differences in lung or liver fibrosis among groups. Fibrosis may occur at a later stage in the mouse model of MPS II or the staining and analysis protocol employed in this study was not sensitive to detect compartmentalized areas of fibrosis that may occur with disease development.

## Conclusions

The findings indicate that the administration of additional idursulfase by nebulization in combination with intravenous delivery augments the long-term retention of active enzyme in the liver but not the lung. The evidence from the liver suggests that active enzymes can be delivered effectively by nebulization and is sequestered in the liver. The finding that lung IDS activity assessed seven days after final treatment was not elevated in the *KO-IV* group relative to the *KO* control may provide insight into respiratory failure remaining the leading cause of death for patients with MPS II [11]. Both treated groups, however, demonstrated lower abundance of enzymes downstream of IDS that are involved in GAG degradation that closely resembled the pattern of *WT* control. These findings suggest that although the enzyme does not persist over the time frame measured, there is a molecular signature showing beneficial effects on biochemical processes of MPS II in the lung.

## Supporting information

**S1 File. Liver proteomics data output from proteome discoverer.**
(ZIP)

**S2 File. Lung proteomics data output from proteome discoverer.**
(ZIP)

## Acknowledgments

The authors thank Dr. Kari Basso, Director of the University of Florida Mass Spectrometry Research and Education Center, and her team at the UF Proteomics and Mass Spectrometry core, for their assistance with the collection and analysis of proteomics data for this study. The authors also extend thanks to the UF Molecular Pathology core for their assistance with the histological staining and imaging used in this study. We are also grateful to Dr. Terence Ryan for assuming administrative responsibilities to allow the authors to complete the study

## Author contributions

**Conceptualization:** Anatalia Labilloy.

**Data curation:** Alex J. Shamoun, Malaica Ashley.

**Formal analysis:** Alex J. Shamoun, Gisienne Reis, Malaica Ashley, Leonardo F. Ferreira.

**Funding acquisition:** Anatalia Labilloy, Leonardo F. Ferreira.

**Investigation:** Alex J. Shamoun, Gisienne Reis, Malaica Ashley, Anatalia Labilloy.

**Methodology:** Alex J. Shamoun, Gisienne Reis, Malaica Ashley, Anatalia Labilloy, Leonardo F. Ferreira.

Project administration: Gisienne Reis, Anatalia Labilloy, Leonardo F. Ferreira.

Resources: Anatalia Labilloy.

Supervision: Anatalia Labilloy, Leonardo F. Ferreira.

Validation: Alex J. Shamoun, Anatalia Labilloy.

Visualization: Alex J. Shamoun, Leonardo F. Ferreira.

Writing – original draft: Alex J. Shamoun.

Writing – review & editing: Gisienne Reis, Malaica Ashley, Anatalia Labilloy, Leonardo F. Ferreira.

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
