## [Decision Letter · Decision Letter 0]

Dear Dr. Ferreira,

We look forward to receiving your revised manuscript.

Kind regards,

Jordan Robin Yaron, Ph.D.

Academic Editor

PLOS ONE

Journal Requirements:

2. To comply with PLOS ONE submissions requirements, in your Methods section, please provide additional information regarding the experiments involving animals and ensure you have included details on (1) methods of sacrifice and (2) efforts to alleviate suffering.

“This study was supported by an investigator-initiated grant from Takeda Pharmaceuticals (IIR-USA-001523) to AL”

4. Please note that funding information should not appear in the Acknowledgments section or other areas of your manuscript. We will only publish funding information present in the Funding Statement section of the online submission form. Please remove any funding-related text from the manuscript. 

**Additional Editor Comments:**

In your resubmission, specifically address the following:

Relationship of the findings of the study, negative as they may be, with potential translation to humans.The percentage of drug which becomes available to the lungs, and contrast this with the effects seen in the liver.What, if any, human data exist in the literature or publicly available databases to provide supplemental support this manuscript.To what degree the proteomics analyses (pathway enrichments) explain the observations made.

Reviewers' comments:

Reviewer's Responses to Questions

**Comments to the Author**

1. Is the manuscript technically sound, and do the data support the conclusions?

Reviewer #1: Yes

Reviewer #2: Yes

2. Has the statistical analysis been performed appropriately and rigorously?

Reviewer #1: Yes

Reviewer #2: Yes

3. Have the authors made all data underlying the findings in their manuscript fully available?

Reviewer #1: Yes

Reviewer #2: Yes

4. Is the manuscript presented in an intelligible fashion and written in standard English?

Reviewer #1: Yes

Reviewer #2: Yes

Reviewer #1: This is a well performed study investigating IDS-KO mice with idursulfase by combination of IV and nebulised form. It is difficult for me to directly comment on the methodology of inhalation & measurements of concentration in the mice as I have not performed such work before.

With regard to the key concepts and manuscript itself, I think it is an interesting and important study. Do you think that this directly extrapolates to humans?

What % of this drug is absorbed into the lung using the nebuliser? It would be important to know if nebulising this drugs in humans would cause bronchoconstriction - can you tell us if there is any data in this area, and add a sentence on this in the conclusion.

You have listed in the conclusion that you were unable to evaluate for any evidence of lung fibrosis. Is there any human data in this area? How do we know that improving activity in the liver and improving the molecular signature in the lungs will have a meaningful effect on patients?

Reviewer #2: In the manuscript by Shamoun et al., the authors tested whether IDS administration using a nebulizer can increase IDS dosage in the lung in IDS-KO mice. However, there was no increase in IDS activity in the lung. The authors also perform proteomics analysis for GO and protein network using KEGG databases and identified some pathways

Although the authors' findings are not significant at all, this should not be a scope of this journal. The presented experiments are performed adequately, although the authors' analysis of the proteomics data remains shallow and limited. also, it was unclear whether the identified pathways by the proteomics analysis can explain the observations exactly. However, this is still acceptable because there may not be significance to perform proteomics analysis when there is no IDS increase in the lung with nebulization.

The followings are minor comments.

Line 75 "multiple respiratory markers with ERT": explain "ERT" beforehand - i.e. describe "enzyme replacement therapy (ERT)" in Line 69

Line 88 "Male wildtype C57BL/6": describe the substrain, C57BL/6N or C57BL/6J?

Line 246: add "fibrosis judged by Masson's Trichrome staining"

Line 247: explain what "percentage area stained for alcian blue" means

All figures for the proteomics data: use larger fonts for readers

**Do you want your identity to be public for this peer review?** For information about this choice, including consent withdrawal, please see our Privacy Policy

Reviewer #1: No

Reviewer #2: No

---

## [Author Response · Author response to Decision Letter 1]

25 Mar 2025

We thank the reviewers and editors for the comments and suggestions to improve the manuscript. We have included a point-by-point response in the file submitted with the manuscript

---

## [Decision Letter · Decision Letter 1]

Nebulized and intravenous enzyme replacement therapy in mice with mucopolysaccharidosis type II

PONE-D-24-38268R1

Dear Dr. Ferreira,

We’re pleased to inform you that your manuscript has been judged scientifically suitable for publication and will be formally accepted for publication once it meets all outstanding technical requirements.

Kind regards,

Jordan Robin Yaron, Ph.D.

Academic Editor

PLOS ONE

Additional Editor Comments (optional):

Reviewers' comments:

Reviewer's Responses to Questions

**Comments to the Author**

Reviewer #2: All comments have been addressed

Reviewer #3: All comments have been addressed

2. Is the manuscript technically sound, and do the data support the conclusions?

Reviewer #2: Yes

Reviewer #3: Yes

3. Has the statistical analysis been performed appropriately and rigorously?

Reviewer #2: Yes

Reviewer #3: Yes

4. Have the authors made all data underlying the findings in their manuscript fully available?

Reviewer #2: (No Response)

Reviewer #3: No

5. Is the manuscript presented in an intelligible fashion and written in standard English?

Reviewer #2: Yes

Reviewer #3: Yes

Reviewer #2: (No Response)

Reviewer #3: no significant increase in lung IDS activity was observed seven days post-treatment in either treatment group which actually deviates the study from the hypothesis.

Because there is no inhalation dosage validation, the amount of IDS administered via nebulization cannot be measured. This restricts the interpretation of comparative efficacy and adds a significant confounding factor.

The combination therapy group received a larger total dose than the IV-only group, as mentioned under their own limitations. This could account for greater liver retention rather than route-dependent distribution.

**Do you want your identity to be public for this peer review?** For information about this choice, including consent withdrawal, please see our Privacy Policy

Reviewer #2: No

Reviewer #3: No

---

## [Editor Report · Acceptance letter]

PONE-D-24-38268R1

PLOS ONE

Dear Dr. Ferreira,

I'm pleased to inform you that your manuscript has been deemed suitable for publication in PLOS ONE. Congratulations! Your manuscript is now being handed over to our production team.

Kind regards,

on behalf of

Dr. Jordan Robin Yaron

Academic Editor

PLOS ONE